**Data Availability Statement:** All relevant data are within the paper and its Supporting information files.

# Does a humoral correlate of protection exist for SARS-CoV-2? A systematic review

**Julie Perry[1,2], Selma Osman[1], James Wright[1], Melissa Richard-Greenblatt[1], Sarah A. Buchan[1,3,4], Manish Sadarangani[5,6], Shelly Bolotin****[1,3,4,7]** *

1 Public Health Ontario, Toronto, Ontario, Canada, 2 Faculty of Dentistry, University of Toronto, Toronto, Ontario, Canada, 3 Dalla Lana School of Public Health, University of Toronto, Toronto, Ontario, Canada, 4 Centre for Vaccine Preventable Diseases, University of Toronto, Toronto, Ontario, Canada, 5 Department of Pediatrics, University of British Columbia, Vancouver, British Columbia, Canada, 6 Vaccine Evaluation Center, BC Children's Hospital Research Institute, Vancouver, British Columbia, Canada, 7 Laboratory Medicine and Pathobiology, University of Toronto, Toronto, Ontario, Canada

* shelly.bolotin@oahpp.ca

## Abstract

### Background

A correlate of protection (CoP) is an immunological marker associated with protection against infection. Despite an urgent need, a CoP for SARS-CoV-2 is currently undefined.

### Objectives

Our objective was to review the evidence for a humoral correlate of protection for SARS-CoV-2, including variants of concern.

### Methods

We searched OVID MEDLINE, EMBASE, Global Health, Biosis Previews and Scopus to January 4, 2022 and pre-prints (using NIH iSearch COVID-19 portfolio) to December 31, 2021, for studies describing SARS-CoV-2 re-infection or breakthrough infection with associated antibody measures. Two reviewers independently extracted study data and performed quality assessment.

### Results

Twenty-five studies were included in our systematic review. Two studies examined the correlation of antibody levels to VE, and reported values from 48.5% to 94.2%. Similarly, several studies found an inverse relationship between antibody levels and infection incidence, risk, or viral load, suggesting that both humoral immunity and other immune components contribute to protection. However, individual level data suggest infection can still occur in the presence of high levels of antibodies. Two studies estimated a quantitative CoP: for Ancestral SARS-CoV-2, these included 154 (95% confidence interval (CI) 42, 559) anti-S binding antibody units/mL (BAU/mL), and 28.6% (95% CI 19.2, 29.2%) of the mean convalescent antibody level following infection. One study reported a CoP for the Alpha (B.1.1.7) variant of concern of 171 (95% CI 57, 519) BAU/mL. No studies have yet reported an Omicron-specific CoP.

**Funding:** This work was supported by Public Health Ontario and funding from the Public Health Agency of Canada. the funders had no role in study design, data collection and analysis, decision to publish, or preparation of the manuscript.

**Competing interests:** The other authors have declared that no competing interests exist.

## Conclusions

Our review suggests that a SARS-CoV-2 CoP is likely relative, where higher antibody levels decrease the risk of infection, but do not eliminate it completely. More work is urgently needed in this area to establish a SARS-CoV-2 CoP and guide policy as the pandemic continues.

## Introduction

Both previous infection and vaccination against SARS-CoV-2 provide protection against infection and severe disease, but the mechanism and durability of that protection remains unclear [1]. Immunity to SARS-CoV-2 is likely both humoral and cellular [2], but it is uncertain whether a correlate of protection (CoP) for SARS-CoV-2 exists, and if so, whether it is easily quantifiable using diagnostic testing. Without a CoP, serological testing cannot confirm immunity, leaving an evidence gap in public health policy particularly as new variants of concern emerge.

A CoP is an immunological marker associated with protection from an infectious agent following infection or vaccination [3]. Some CoPs are mechanistic (i.e. directly responsible for protection), while others are non-mechanistic or surrogate, and although not directly responsible for protection, can be used in substitute of the true correlate [3, 4]. A CoP can be absolute, where protection against disease is certain above a threshold, or relative, where higher levels of a biomarker correspond to more protection [2]. Some correlates vary by endpoint (e.g. symptomatic infection or severe disease), or are only applicable to a specific endpoint [3]. The majority of CoPs described are humoral and used in a surrogate manner, as these antibodies are easier to detect in clinical laboratory settings than components of cellular immunity [5].

Elucidating a CoP for SARS-CoV-2 is critical for improving our understanding of the extent and duration of protection against infection for individuals and populations. At the individual level, a CoP would provide clear immunological vaccine trial endpoints, and therefore may provide a pathway to licensure for new vaccines [5]. If measurable using a diagnostic test, a CoP would enable determination of individual and community-level immunity, which is particularly important for immunocompromised individuals [6, 7] and the assessment of population level immunity through serosurveys [5].

The search for a SARS-CoV-2 CoP is further complicated by the emergence of variants of concern (VOCs). Sera from previously infected and/or vaccinated individuals have reduced neutralizing ability against VOCs including Beta (B.1.351), Delta (B.1.617.2) and Omicron (B.1.1.529) [8–10], with the latter showing the greatest extent of immune evasion of all VOCs thus far [11]. This variation raises the possibility that a SARS-CoV-2 CoP may be VOC-specific.

With these facts in mind, and considering that an easily measurable CoP would most likely be humoral and not cellular, we performed a systematic review to assess the evidence for a humoral CoP for SARS-CoV-2.

## Methods

### Data sources and searches

We searched the OVID MEDLINE database from inception to December 31, 2021, and the EMBASE, Global Health, Biosis Previews and Scopus databases from inception to January 4, 2022. We used the NIH iSearch COVID-19 Portfolio tool to search for preprint articles published up to December 31, 2021. Our search included studies reporting either re-infection or breakthrough infection following vaccination. Full search terms used are reported in S1 Table.

We also searched reference lists for suitable articles, and requested article recommendations from experts in the field.

## Study selection

One reviewer screened titles and abstracts using Distiller SR (Ottawa, Ontario, Canada). Studies passed title and abstract screening if their abstracts discussed re-infection with SARS-CoV-2 or breakthrough infection following vaccination; mentioned antibody measures specific to SARS-CoV-2; or mentioned a correlate or threshold of protection against SARS-CoV-2. We excluded studies that focused on immunocompromised populations or animal models.

Two reviewers screened full texts of articles that passed title/abstract screening using defined criteria (Table 1). We included studies reporting a quantitative CoP against SARS-CoV-2, and studies reporting re-infection or breakthrough infection along with associated pre-infection measures. If these studies reported aggregate antibody measures (i.e. geometric mean titres (GMT)) we required them to include summary statistics (i.e. statistical significance testing or 95% confidence intervals (95% CI)). We also included studies that correlated antibody levels to vaccine efficacy (VE) or effectiveness, but only if they provided statistical summary measures (e.g. a correlation co-efficient describing the relationship between antibody level and VE), or if they correlated an antibody concentration to a VE of 100% (i.e. absolute protection). We only included studies written English or French. We calculated a Cohen's Kappa to assess inter-rater agreement for full-text screening.

## Data extraction and quality assessment

Two reviewers extracted data in duplicate from articles that met full-text screening criteria, using WebPlotDigitizer [12] for figures. We used the National Institutes of Health National Heart, Lung and Blood Institute (NIH NHLBI) Study Quality Assessment tools to assess study quality [13], adapting it by adding questions specific to this study. Studies correlating VE to antibody levels were evaluated using the Cohort and Cross Sectional Tool.

**Table 1. Definitions applied to determine cases of re-infection and breakthrough in this systematic review.**

| Term | Definition |
| --- | --- |
| SARS-CoV-2 re-infection, suspected case | A symptomatic person with a positive molecular test result for SARS-CoV-2 following a period of ≥45 days from the first infection with SARS-CoV-2, or An asymptomatic person with a positive molecular test result for SARS-CoV-2 following a period ≥90 days from the first infection with SARS-CoV-2, for which SARS-CoV-2 shedding from a previous infection, or an infection of a different etiology have been ruled out [55]. |
| SARS-CoV-2 re-infection, confirmed case | A person who meets the suspected case criteria, but also has a documented time interval for which they were not symptomatic, did not shed SARS-CoV-2 virus or RNA, or had a negative SARS-CoV-2 laboratory test. In addition, the case has had whole genomic sequencing of both the initial and subsequent SARS-CoV-2 virus, with evidence that they belong to different clades or lineages or exhibiting a number of single nucleotide variations that correlate with the probability that each virus is from a different lineage [55]. |
| SARS-CoV-2 breakthrough infection with one vaccine dose | A positive molecular test result in an individual who received one dose of a vaccine product that is approved in at least one jurisdiction (i.e.–not an experimental vaccine) at least 14 days previously [56]. |
| SARS-CoV-2 breakthrough infection with two vaccine dose | A positive case molecular test result in an individual who received a second dose of a vaccine product that is approved in at least one jurisdiction (i.e.–not an experimental vaccine) at least seven days previously [57] |

### Data synthesis and analysis

We reported our results using the Preferred Reporting Items for Systematic Reviews and Meta-Analyses (PRISMA) 2020 [14]. A PRISMA reporting checklist can be found in the Supplemental files section (S1 Checklist). Recognizing that that the immune response following natural infection and vaccination may differ, we grouped studies involving re-infection separately from studies examining breakthrough infection.

## Results

We identified 11,803 records for screening (Fig 1). After de-duplication, we screened 4,919 peer-reviewed studies, 783 preprint studies and 16 studies identified through expert recommendations and scanning of article reference lists. After title/abstract screening, full-text screening (Kappa = 1.0) and quality assessment, we included 25 articles in our review. Of these, 14 described SARS-CoV-2 re-infection along with individual or aggregate humoral measures [15–28], and 11 studies described SARS-CoV-2 breakthrough infection following vaccination or statistical modelling to explore associations between VE and antibody levels [29–39] (Table 2). Only two studies estimated a SARS-CoV-2 antibody CoP, both using statistical modelling methods [33, 34].

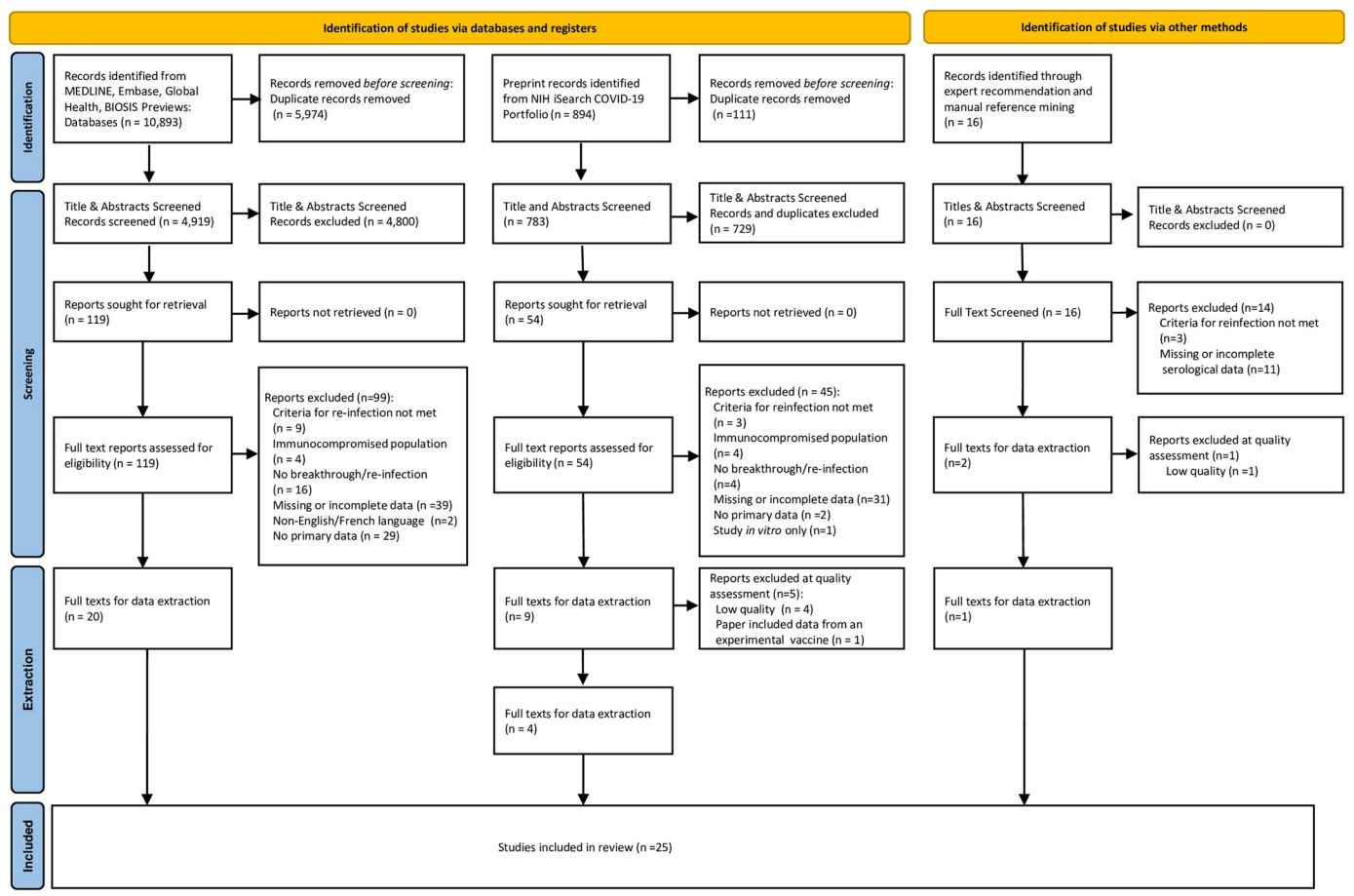

**Fig 1. PRISMA diagram.**

**Table 2. Summary of articles included in this review following re-infection and breakthrough infection definition screening, and types of evidence they describe.**

| Evidence | Included articles | Number of articles |
|---|---|---|
| SARS-CoV-2 re-infection<br>• Describing individual or aggregate humoral measures | Dimeglio et al. [17], Roy et al. [23], Krukitov et al. [20], Leidi et al. [21], Ul-Haq et al. [25], Vetter et al. [26], Ali et al. [15], Gallais et al. [18], Brehm et al. [16], Inada et al. [19], Selhorst et al. [24], Wilkins et al. [27], Lumley et al. [22], Munivenkatappa et al. [28] | 14 |
| SARS-CoV-2 breakthrough infections following vaccination<br>• Describing individual or aggregate humoral measures | Strafella et al. [37], Schulte et al. [36], Michos et al. [35], Bergwerk et al. [29], Feng et al. [39], Yamamoto et al. [38] | 11 |
| • Describing statistical modelling to explore associations between VE and antibody levels | Khoury et al. [34], Earle et al. [31], Goldblatt et al. [33], Cromer et al. [30] | |
| • Describing both aggregate humoral measures and statistical modelling to explore associations between VE and antibody levels | Gilbert et al. [32] | |
| Total | | 25 |

## Studies describing SARS-CoV-2 re-infection

Fourteen studies met our SARS-CoV-2 re-infection definition and provided pre-infection antibody values (Table 3). These included seven cohort studies [15, 17, 18, 20–22, 27], and seven case reports [16, 19, 23–26, 28]. The majority of studies reported re-infection in health-care workers, patients, or long term care home residents [15–18, 20, 22, 24–28], with a minority reporting re-infection in the general population [19, 21, 23]. When reported, specimen collection occurred between 14 days and seven months after initial infection [16, 26] and between 4 days and seven months before re-infection [20, 27]. Antibody testing methods included various commercial and laboratory developed enzyme-linked immunosorbent assays (ELISAs) targeting anti-spike (anti-S), anti-receptor binding domain (anti-RBD) and anti-nucleocapsid (anti-N) antibodies, as well as neutralization assays. No study utilized the World Health Organization (WHO) International Standard (IS) [40]. Only three papers reported on the SARS-CoV-2 lineage of the re-infection [16, 24, 26], with none reporting serological measures preceding re-infection with VOCs.

Two studies compared antibody levels between re-infected and protected individuals. Krutikov et al. found no statistically significant difference in anti-N IgG between cases and controls (p = 0.544) but showed that individuals who were antibody-negative at baseline were at greater risk of infection than those who were antibody-positive [20]. Using Poisson regression, Lumley and colleagues also found that anti-S positive individuals were less likely to be infected compared to those who were anti-S negative (incidence rate ratio (IRR) of 0.11 (95% CI 0.03, 0.44)) [22]. Similar findings were observed using anti-N antibody (IRR = 0.11 (95% CI 0.03, 0.45)). Analysis of the association between continuous antibody concentrations and incidence was also statistically significant for both antibodies (p<0.001) [22].

## Studies reporting antibody measures related to breakthrough infection or VE

We included 11 studies describing breakthrough SARS-CoV-2 infection. These included two case reports [36, 37], one cohort study [35], two case-control studies [29, 38], and two studies that re-analyzed antibody data from a clinical trial [32, 39]. Five in silico studies utilized

**Table 3. Articles describing SARS-CoV-2 re-infection along with individual or aggregate humoral measures[#].**

| First author, publication year (study country) | Study design, population | Number of reinfections reported | Lineage of first infection, reinfection | Time from first infection to most recent antibody test before re-infection* (days) | Antibody assay, target isotype (cut-off) | Pre reinfection antibody level* | Time from most recent antibody test* to re-infection (days) | Statistical association |
|---|---|---|---|---|---|---|---|---|
| **Inada, 2020 (Japan)** | Case report, general public | 1 | Not provided | 94 | Laboratory developed Anti-S IgG ELISA (cut-off not provided) | 15.6 OD ratio | 11 | None reported |
| | | | | 94 | Laboratory developed neutralization assay, IgG specific | 50 µg/mL | 11 | None reported |
| **Roy, 2021 (Not Reported)** | Case report, general public | 1 | Not provided | 150 (5 months) | LIASON SARS-CoV-2 S1/S2 IgG test kit (DiaSorin Inc., Saluggia, Italy) (>15.0) | 48 AU/ml | 47 | None reported |
| **Dimeglio, 2021 (France)** | Cohort, HCW | 5 | Not provided | Not provided | Quantitative ELISA (Wantai Biological Pharmacy Enterprise Co, Ltd, China); Total Ab; anti-Spike | Range: 1.5–385.8 S/Co | Not provided (serology performed a median of 167 IQR (156–172) days apart) | None reported |
| | | | | Not provided | Neutralization test—assay not provided | Range: 0–64 S/CO | Not provided (serology performed a median of 167 days apart) | None reported |
| **Leidi, 2021 (Switzerland)** | Cohort, general public | 5 | Not provided | Not provided | Euroimmun ELISA, (Euroimmun Lubeck, Germany); IgG; anti-S (cut-off: ≥0.5) | Range: 0.58–2 ratio | Range: 34–185 | None reported |
| **Lumley, 2021 (England)** | Cohort, HCW | 3 | Not provided | 50–112 days for HCW2; Not provided for HCW1 and HCW3 | ELISA (LDT); IgG; Anti-S (cut-off not provided) | Range: 0.34–10.5 million units | Range: 61–179 | IRR of 0.11 (95% CI 0.03, 0.44, p = 0.002) in seropositive healthcare workers compared to seronegative healthcare workers |
| | | | | 50–112 days for HCW2; Not provided for HCW1 and HCW3 | ELISA (LDT); IgG; Anti-N (cut-off not provided) | Range: 0–7.5 arbitrary units | Range: 10–179 | IRR of 0.11 (95% CI 0.03, 0.45, p = 0.002) in seropositive healthcare workers compared to seronegative healthcare workers |
| **Ul-Haq, 2020 (Pakistan)** | Case report, HCW | 1 | Not provided | 15 | Assay information not provided, cut off of ≥1 | 1.97 | 133 | None reported |

*(Continued)*

**Table 3.** (*Continued*)

| First author, publication year (study country) | Study design, population | Number of reinfections reported | Lineage of first infection, reinfection | Time from first infection to most recent antibody test before re-infection* (days) | Antibody assay, target isotype (cut-off) | Pre reinfection antibody level* | Time from most recent antibody test* to re-infection (days) | Statistical association |
|---|---|---|---|---|---|---|---|---|
| **Vetter, 2021 (Switzerland)** | Case report, HCW | 1 | Re-infection lineage different than first infection, but both clade 20A | 35 | Euroimmun Anti-S IgG (Euroimmun, Lubeck, Germany) (cut-off not provided) | 2.16 UI/l | 169 | None reported |
| | | | | 35 | Elecsys/Roche (Basel, Switzerland), Total anti-RBD (0.8 U/ml) | 21.6 U/ml | 169 | |
| | | | | 35 | Elecsys/Roche (Basel, Switzerland), Total anti-N (cut-off not provided) | 128 COI | 169 | |
| | | | | 35 | PRNT/neutralization assay 90% | 14.1 (1/) (inferred to mean 1/14.1) | 169 | |
| **Ali, 2020 (Iraq)** | Cohort, patients admitted to hospital | 17** | Not provided | Not provided | IgG Anti-N (PishTaz Teb Diagnostic, Tehran, Iran) (cut-off = 1.1) | 5.87 (s/ca) | Not provided | None reported |
| **Gallais, 2021 (France)** | Cohort, HCW | 1 | Not provided | 96 | Abbott Architect SARS-CoV-2 IgG Quant II assay (Abbott, Sligo, Ireland) (cut-off:50AU/ml) | 2.6 log AU/ml | 7 months (number of days not reported) | None reported |
| | | | | 96 | EDI Novel coronavirus COVID-19 IgG ELISA (San Diego, USA) (no cut-off reported) | 1.0 OD S/CO | 7 months (number of days not reported) | |
| **Brehm, 2021 (Germany)** | Case report, HCW | 1 | B.3, B.1.177 | ~6 months | Diasorin IgG Anti-S (Saluggia, Italy) (cut-off: 15 AU/mL) | 60 AU/mL | ~4 months (number of days not reported) | None reported |
| | | | | 210 | Indirect immunofluorescence, IgG, IgM, IgA | IgG 1:320 | 73 | |
| | | | | | | IgM <1:20 | | |
| | | | | | | IgA <1:20 | | |
| | | | | 210 | Neutralization Assay | Local Hamburg reference isolate (HH-1): | 73 | |
| | | | | | | 1:80 IC50 | | |
| | | | | | | B.1.177: 1:160 IC50 | | |
| **Selhorst, 2020 (Belgium)** | Case report, HCW | 1 | V clade, G clade | 105 | Roche Total anti-N (Basel, Switzerland) (cut-off: ≥1) | 102 cut-off/ index | 80 | None reported |
| | | | | 94 | PRNT/neutralization assay; 2019-nCoV-Italy-INMI1; NT50 | $NT_{50}$ 200 | 91 | |

(*Continued*)

**Table 3.** (Continued)

| First author, publication year (study country) | Study design, population | Number of reinfections reported | Lineage of first infection, reinfection | Time from first infection to most recent antibody test before re-infection* (days) | Antibody assay, target isotype (cut-off) | Pre reinfection antibody level* | Time from most recent antibody test* to re-infection (days) | Statistical association |
|---|---|---|---|---|---|---|---|---|
| **Munivenkatappa, 2021 (India)** | Case report, HCW | 1 | Not provided | 76 days | ELISA (LDT), IgG, anti-RBD (no cut-off provided) | Ratio of positive to negative: 4.14 | 31 days | None reported |
| | | | | 76 days | ELISA (LDT), IgG, anti-N (no cut-off provided) | Ratio of positive to negative: 8.57 | 31 days | None reported |
| | | | | 76 days | PRNT/Neutralization assay, no details provided | Positive (no quantitative result given) | 31 days | |
| **Krutikov, 2021 (England)** | Cohort, staff and residents in LTC | 14 | Not provided | Not provided | Mesoscale Diagnostics (MSD) IgG, anti-S (Rockville, USA) (no cut-off provided) | Range: 78–137840 AU/mL | Range: 12–132 | Cox regression showed antibody-negative staff and residents at baseline had increased risk of PCR+ infection than those antibody-positive at baseline (aHR range: 0.08 (95% CI 0.03, 0.23) -0.39 (95% CI 0.19, 0.82)) |
| | | | | Not provided | Mesoscale Diagnostics (MSD) IgG, anti-N (Rockville, USA) (no cut-off provided) | Range: 137–222308 AU/ml; Median antibody levels of 101527 (95% CI 18393, 161580) AU/mL for cases, and 26326 (95% CI 14378, 59633) AU/mL for controls. | Range: 12–132 | No statistically significant difference between antibody levels of individuals re-infected and those not (p = 0.544) |
| **Wilkins, 2021 (USA)** | Cohort study, HCW | 8 | Not provided | Not provided | Abbott ARCHITECT i2000SR Immunoassay system, IgG, anti-N (Sligo, Ireland) (cut-off: ≥1.4) | Range: 1.92–6.01 Index Value | Range: 95–212 | None reported |

#—Assay results from each study were included for every antibody type (i.e.–anti-S, anti-N, anti-RBD) and isotype (i.e.–IgG, IgM, IgA) measured. In instances where more than one assay target was used to measure the same antibody target in the same study (i.e.–both PRNT and pseudoneutralization results, or anti-S results from two different assays), we included only one of these results. Full data extraction for every study can be provided upon request.

*- if more than one test result was provided, the result closest in time to re-infection is presented.

**—In these studies, other reinfections were reported as well, but with no accompanying temporal and laboratory data, or did not met our reinfection criteria

Definitions: anti-S = anti-spike, anti-N = anti-nucleocapsid, anti-RBD = anti-receptor binding domain, PRNT = plaque reduction neutralization test, LDT = laboratory-developed test, ELISA = enzyme-linked immunosorbent assay, AU = arbitrary units, OD = optical density, IRR = increased relative risk, HCW = health care worker, LTC = long term care

statistical methods to explore the association between antibody levels and VE [30–34]. The populations studied were either clinical trials or other vaccine study participants [30–34, 39] or healthcare workers [29, 35–38]. Three studies reported results in WHO IS units (binding antibody units (BAU)/mL) [32, 33, 37].

Of the 11 studies describing breakthrough infection, six studies provided individual or aggregate humoral measures [29, 35–39], four studies used statistical modelling to explore associations between VE and antibody levels [30, 31, 33, 34], and one study included both humoral measures and statistical modelling [32] (Tables 4 and 5). Five studies [29, 36–39] reported the lineage of the breakthrough infection, and two modeling studies include VOCs in their analysis [30, 33].

**Studies describing breakthrough infections following SARS-CoV-2 vaccination.** Seven of 11 studies reported antibody levels following one [35] or two doses of COVID-19 vaccine, including BNT162b2 (Pfizer-BioNTech) [29, 35–38] mRNA-1273 (Moderna) [32] and ChA-dOx1 nCoV-19 (AstraZeneca) [39] (Table 4). Sera were collected between nine [36] and 109 days [32] after the second vaccine dose, but the time from sampling to breakthrough infection was not always reported. Antibody levels were assessed using a variety of commercial serology assays and/or neutralization assays. Five studies reported the viral lineage, including three studies reporting Alpha (B.1.1.7) [29, 37, 39], one reporting B.1.525 [36], and one reporting Delta (B.1.617.2) [38] infections.

Four studies compared aggregate antibody levels between cases and non-cases. Gilbert et al. calculated geometric mean concentration (GMC) ratios of cases to non-cases, ranging from 0.57 (95% CI 0.39, 0.84) to 0.71 (95% CI 0.54, 0.94), depending on antibody target and sampling interval [32]. Using Cox regression, the authors found statistically significant associations between increasing antibody levels and decreasing risk of COVID-19. Bergwerk et al. applied generalizing estimating equations to predict antibody levels and generate GMT ratios of cases to non-cases. For neutralizing antibodies, these ranged from a case-to-control ratio of 0.15 (95% CI, 0.04, 0.55) within the first month after the second vaccine dose to 0.36 (95% CI 0.17, 0.79) by the week before breakthrough infection [29]. Using linear regression, this study demonstrated a statistically significant correlation between cycle threshold (Ct) value of cases and neutralizing antibody level, suggesting an inverse relationship between antibody level and viral load. Feng and colleagues found no statistically significant difference between median antibody levels of cases and non-cases [39]. However, using a generalized additive model, symptomatic infection risk was found to be inversely correlated to antibody levels. Yamamoto et al. found no statistically significant difference in post-vaccination neutralization levels in healthcare workers who experienced a breakthrough infection and matched controls during the Delta wave [38]. The authors found that neutralizing titres were lower against Alpha and Delta variants than the wild-type virus, but were comparable between cases and controls.

**Studies reporting associations between antibody levels and VE.** Five studies described correlations between antibody levels and VE against BNT162b2 [30, 31, 33, 34], mRNA-1273 [31–34], ChAdOx1 nCoV-19 [30, 31, 33, 34], Ad26.COV2.S (Janssen/ Johnson and Johnson) [30, 31, 33, 34], NVX-CoV2373 (Novavax) [30, 31, 34], CoronaVac (SinoVac) [31, 34], and rAd26+S+rAd5-S (Gamaleya Research Institute) [31, 34] vaccine using re-analyzed clinical trial and other vaccine. The studies generated correlations using either neutralizing antibody levels, derived through plaque reduction neutralization tests (PRNT) or microneutralization assays, or IgG levels measured through ELISAs.

Three of five studies [30, 31, 34] reported correlation coefficients for the relationship between neutralizing antibodies and VE ranging from 0.79 to 0.96. Two studies [31, 33] reported correlation coefficients of 0.82 to 0.94 to describe the relationship between anti-Spike IgG and VE. Since serology and neutralization assays were not calibrated to a common

**Table 4. Articles describing breakthrough following SARS-CoV-2 infection along with individual or aggregate humoral measures[#].**

| First author, publication year (study country) | Study design, population | Vaccines included in study and number of doses | Number of cases reported | Lineage of breakthrough infection | Time from last vaccine dose to antibody test[*] (days) | Antibody assay and target, isotype (cut-off) | Pre-breakthrough antibody level[*] | Time from antibody test[*] to breakthrough infection (days) | Statistical association |
|---|---|---|---|---|---|---|---|---|---|
| **Strafella, 2021 (Italy)** | Case report, HCW | Pfizer, 2 doses | 1 | B.1.1.7 | 26 | Euroimmun Anti-Sars-CoV-2, IgG Anti-S1, IgA Anti-S1, IgM Anti-N (Lubeck, Germany) (cut-off: ≥1.1) | IgG: 10.47 ratio units / IgA: 3.58 ratio units / IgM: 0.2 ratio units | 26 | None reported |
| | | | | | 26 | Roche Elecsys Anti-Sars-CoV-2 Total anti-RBD (Basel, Switzerland) (cut-off: >0.8 BAU/ml) | 978.7 U/ml | 26 | None reported |
| **Schulte, 2021 (Germany)** | Case report, HCW | Pfizer, 2 doses | 1** | B.1.525 | 9 | Roche, Total Ig, S1 (Basel, Switzerland) (cut-off not provided) | >250 U/mL | 45 | None reported |
| **Gilbert, 2021 (USA)** (Please see Table 5 for additional evidence) | Nested case-cohort within an RCT, vaccine trial participants | Moderna, 2 doses | 55 (text) or 46 (Table 1) | Not provided | ≤81 | MSD anti-S, IgG (Rockville, USA) (cut-off: >10.8424 IU/mL) | GMC of 1890 (95% CI 1449, 2465) IU/mL among cases, 2652 (95% CI 2457, 2863) IU/mL among non-cases. | Not provided | GMC ratio of cases/non-cases = 0.71 (95% CI 0.54, 0.94) / Cox regression to estimate association between risk of COVID-19 and anti-S IgG level (per 10-fold increase). HR = 0.66 (95% CI 0.50, 0.88). / 34% decrease in risk for every 10-fold increase of Anti-S IgG |
| | | | | | ≤81 | MSD anti-RBD, IgG (Rockville, USA)(cut-off: >14.0858 IU/mL) | GMC of 2744 (95% CI 2056, 3664) IU/mL among cases, 3937 (95% CI 3668, 4227) IU/mL among non-cases | Not provided | GMC ratio of cases/non-cases 0.70 (95% CI 0.52, 0.94) / Cox regression to estimate association between risk of COVID-19 and anti-RBD IgG level (per 10-fold increase). HR = 0.57 (95% CI 0.40, 0.82). / 43% decrease in risk for every 10-fold increase of Anti-RBD IgG |
| | | | | | ≤81 | Pseudoneutralization assay with ID50 calibrated against WHO IS, neutralizing antibodies (no cut-off reported) | GMT of 160 (95% CI 117, 220) ID50 titre among cases, 247 (95% CI 231, 264) ID50 titre among non-cases. | Not provided | GMT ratio of cases/non-cases = 0.65 (95% CI 0.47–0.90) / Cox regression to estimate association between risk of COVID-19 and neutralizing antibody level (per 10-fold increase). HR = 0.42 (95% CI 0.27, 0.65). / 58% decrease in risk for every 10-fold increase of neutralizing antibodies |
| | | | | | | Pseudoneutralization assay with ID80 calibrated against WHO IS, neutralizing antibodies (no cut-off reported) | GMT of 332 (95% CI 248, 444) ID80 titre among cases, 478 (95% CI 450, 508) ID80 titre among non-cases. | | GMT ratio of cases/non-cases = 0.69 (95% CI 0.52, 0.93) / Cox regression to estimate association between risk of COVID-19 and neutralizing antibody level (per 10-fold increase). / HR = 0.35 (95% CI 0.20, 0.61). / 65% decrease in risk for every 10-fold increase of neutralizing antibodies |

*(Continued)*

**Table 4.** (Continued)

| First author, publication year (study country) | Study design, population | Vaccines included in study and number of doses | Number of cases reported | Lineage of breakthrough infection | Time from last vaccine dose to antibody test* (days) | Antibody assay and target, isotype (cut-off) | Pre- breakthrough antibody level* | Time from antibody test* to breakthrough infection (days) | Statistical association |
|---|---|---|---|---|---|---|---|---|---|
| **Feng, 2021 (UK)** | Cohort study secondary analysis of clinical trial data | AstraZeneca | 171** | Mostly B.1.1.7 and B.1.177 | 14–42 | MSD anti-S, IgG, (Rockville, USA) (no cut-off reported) | Median of 30501 (95% CI 16088, 49529) AU/mL for cases, and 33945 (95% CI 18450, 59260) AU/mL for non-cases | Not provided | Generalized additive model to estimate risk of symptomatic COVID-19. |
| | | | | | | | | | Difference between median antibody levels for cases and non-cases: p>0.05 |
| | | | | | | | | | Risk was inversely correlated to anti-spike IgG (p = 0.003), |
| | | | | | | | | | There was no association between risk of asymptomatic COVID-19 and anti-spike IgG |
| | | | | | 14–42 | MSD Anti-RBD, IgG (Rockville, USA) (no cut-off reported) | Median of 40884 (95% CI 20871, 62934) AU/mL for cases, 45693 (95% CI 24009, 82432) AU/mL for non-cases | Not provided | Difference between median antibody levels for cases and non-cases: p>0.05 |
| | | | | | | | | | Risk was inversely correlated to anti-RBD IgG (p = 0.018). |
| | | | | | | | | | There was no association between risk of asymptomatic COVID-19 and anti-RBD IgG |
| | | | | | 14–42 | Microneutralization assay, neutralizing antibodies (no cut-off reported) | Median titre of 206 (95% CI 124, 331) for cases, 184 (95% CI 101, 344) for non-cases | Not provided. Median follow up period of 53 days (IQR 29,81), starting 7 days after blood draw. | Difference between median antibody levels for cases and non-cases: p>0.05 |
| | | | | | | | | | Risk was inversely correlated to microneutralization titre (p<0.001). |
| | | | | | | | | | There was no association between risk of asymptomatic COVID-19 and neutralizing antibodies |
| **Bergwerk, 2021 (Israel)** | Case-control study, HCW | Pfizer, 2 doses | 22** | B.1.1.7 was identified in 85% of breakthroughcases, similar to community prevalence at the time | Median of 36 days (breakthrough infections), median of 35 days (controls) | Beckman Coulter, anti-S1 (Brea, USA)(no cut-off provided) | Case predicted anti-S IgG GMT: 11.2 (95% CI 5.3, 23.9); Control predicted GMT: 21.8 (95% CI 18.6,25.52) | Within a week of breakthrough for cases. Controls were matched to cases by time between second vaccine dose and serology test | Ratio of cases/control GMT: 0.514 (95% CI 0.282, 0.937) |
| | | | | | | | | | Linear regression to assess correlation between Ct value of cases and neutralizing antibody level during peri-infection period. |
| | | | | | | | | | Slope = 171.2 (95% CI 62.9, 279.4). |
| | | | | | Median of 36 days (breakthrough infections), median of 35 days (controls) | Pseudoneutralization assay | Case predicted GMT: 192.8 (95% CI 67.6, 549.8); Control predicted GMT: 533.7 (95% CI 408.1, 698.0) | Within a week of breakthrough for cases. Controls were matched to cases by time between second vaccine dose and serology test | Ratio of cases/control GMT: 0.361 (95% CI 0.165, 0.787) |
| **Michos, 2021 (Greece)** | Cohort study, HCW | Pfizer, 2 doses | 2 | Not provided | One month | GenScript cPass SARS-CoV-2 Neutralization antibody detection kit (Piscataway, USA) | 90 and 95% neutralization | ~10 days | None reported |

*(Continued)*

**Table 4.** (Continued)

| First author, publication year (study country) | Study design, population | Vaccines included in study and number of doses | Number of cases reported | Lineage of breakthrough infection | Time from last vaccine dose to antibody test* (days) | Antibody assay and target, isotype (cut-off) | Pre- breakthrough antibody level* | Time from antibody test* to breakthrough infection (days) | Statistical association |
|---|---|---|---|---|---|---|---|---|---|
| **Yamamoto, 2021 (Japan)** | Case control study, HCW | Pfizer, 2 doses | 17 | 5 of 17 reported to be Delta | Median of 63 (IQR 43–69) days for cases; 62 (IQR 40–69) days for controls | Abbott Advise Dx SARS-CoV-2 IgG II (Sligo, Ireland), anti-RBD, (no cutoff provided) | Case predicted GMC: 5129 (95% CI 3881, 6779); Control predicted GMC: 6274 (95% CI 5017,7847) | 55 (45–64) days | Ratio of cases/control GMC: 0.82 (95% CI 0.65, 1.02), p = 0.07 |
| | | | | | Median of 63 (43–69) days for cases; Median of 62 (40–69) days for controls | Roche Elecsys Anti-SARS-CoV-2 (Basel, Switzerland), Spike total antibody, (no cutoff provided) | Case predicted GMC: 1144 (95% CI 802,1632); Control predicted GMC: 1208 (95% CI 1053–1385) | 55 (45–64) days | Ratio of cases/control GMC: 0.95 (95% CI 0.70, 1.27), p = 0.72 |
| | | | | | Median of 63 (43–69) days for cases; Median of 62 (40–69) days for controls | PRNT/neutralization test (SARS-CoV-2 ancestral, Alpha and Delta strains) | Ancestral strain: case predicted GMT: 405 (95% CI 327,501); Control predicted GMT: 408 (320,520) | 55 (45–64) days | Ratio of cases/control GMT: 0.99 (95% CI 0.74, 1.34), p = 0.96 |
| | | | | | | | Alpha: Case predicted GMT: 116 (95% CI 80,169); Control predicted GMT: 122 (95% CI 96,155) | | Ratio of cases/control GMT: 0.95 (95% CI 0.71, 1.28), p = 0.76 |
| | | | | | | | Delta: Case predicted GMT: 123 (95% CI 85, 177); Control predicted GMT: 135 (95% CI 108, 170) | | Ratio of cases/control GMT: 0.91 (95% CI 0.61, 1.34), p = 0.63 |

# —Assay results from each study were included for every antibody type (i.e.–anti-S, anti-N, anti-RBD) and isotype (i.e.–IgG, IgM, IgA) measured. In instances where more than one assay target was used to measure the same antibody target in the same study (i.e. both PRNT and pseudoneutralization results, or anti-S results from two different assays), we included only one of these results. Full data extraction for every study can be provided upon request.

* - If more than one test result was provided, the result closest in time to re-infection is presented.

** —In these studies, other breakthrough infections were reported as well, but with no accompanying temporal and laboratory data

Definitions: anti-S = anti-spike, anti-N = anti-nucleocapsid, anti-RBD = anti-receptor binding domain, PRNT = plaque reduction neutralization test, LDT = laboratory-determined test, ELISA = enzyme-linked immunosorbent assay, AU = arbitrary units, OD = optical density, IRR = increased relative risk, HCW = health care worker, LTC = long term care, GMC = geometric mean concentration, GMT = geometric mean titre, 95% CI = 95% confidence interval, ID50 = infectious dose titer 50, WHO IS = World Health Organization SARS-CoV-2 antibody International Standard, HR = hazard ratio, RCT = randomized controlled trial, MSD = Mesoscale Discovery

**Table 5. Articles describing statistical modelling to explore associations between VE and antibody levels[#].**

| First author and publication year | Vaccine(s) investigated | Antibody assay and target, isotype | Primary outcome | Correlation | Statistical model used | Result and interpretation | Reported correlate of protection |
|---|---|---|---|---|---|---|---|
| **Earle, 2021** | Pfizer, Moderna, Sputnik, | Neutralization or pseudoneutralization assays, neutralizing antibody<br><br>Results normalized to HCS | PCR confirmed infection, with or without symptomatic illness, or seroconversion measures (varies by study) | Spearman rank ρ = 0.79 | Locally estimated scatterplot smoothing (LOESS) regression, with a tricube weight function | Neutralizing antibody accounted for 77.5% of variation in efficacy | Not provided |
|  | AstraZeneca, Sinovac, Novavax, and Johnson & Johnson | Various ELISAs targeting anti-spike, anti S1 or anti-RBD, IgG<br><br>Results normalized to HCS |  | Spearman rank ρ = 0.93 | Locally estimated scatterplot smoothing (LOESS) regression, with a tricube weight function | Anti-spike IgG accounted for 94.2% of variation in efficacy |  |
| **Khoury, 2021** | Pfizer, Moderna, Sputnik, AstraZeneca, Sinovac, Novavax, and Johnson & Johnson | Various neutralization or microneutralization assays, neutralizing antibody<br><br>Results normalized to HCS | PCR confirmed infection with no symptoms, symptomatic illness, or moderate to severe/critical illness (varies by study) | Spearman's rank ρ = 0.905 | Logistic model | 20.2% (95% CI 14.4, 28.4) of the mean convalescent level estimated to protect 50% of people | Neutralization titre of 1:10 to 1:30, or 54 (95% CI 30, 96) IU/mL |
|  |  |  |  |  | Protective neutralization classification model (a distribution-free approach, using individual neutralization levels)<br>Logistic model | 28.6% (95% CI = 19.2, 29.2%) of the mean convalescent level estimated to provide protection in 100% of people<br><br>3.0% (95% CI 0.71, 13.0) of the mean convalescent level estimated to protect 50% of people against severe disease | 28.6% of mean convalescent level |
| **Cromer, 2021** | Pfizer, AstraZeneca, Novavax, Johnson & Johnson | Neutralization assay (unspecified, reference not included) using Ancestral, Alpha, Beta and Delta strains | Any infection, symptomatic disease, PCR confirmed infection (varies by study) | Spearman's rank ρ = 0.810 | N/A | N/A | Not provided |
| **Goldblatt, 2021** | Pfizer, Moderna, AstraZeneca, Johnson & Johnson | Anti-spike | Antibody threshold at which individual is protected | Spearman's rank ρ = 0.940 | Weighted least squares linear regression | Anti-spike antibodies accounted for 97.4% of the variance in efficacy | Not provided |
|  | Pfizer, Moderna, AstraZeneca, Johnson & Johnson | Anti-spike | Antibody threshold at which individual is protected against Alpha | Spearman's rank ρ = 0.83 | Weighted least squares linear regression | Anti-Spike antibodies accounted for 68.6% of the variation in efficacy | Not provided |
|  | Pfizer, Moderna, AstraZeneca, Johnson & Johnson | Anti-spike | Antibody threshold at which individual is protected |  | Random effects meta-analysis of each vaccine's reverse cumulative distribution function | Individuals with anti-S IgG lab result of at least 154 BAU (95% CI: 42, 559) are protected from infection | Anti-S IgG: 154 BAU (95% CI: 42, 559) |
|  | Pfizer, Moderna, AstraZeneca, Johnson & Johnson | Anti-spike | Antibody threshold at which individual is protected against Alpha |  | Random effects meta-analysis of each vaccine's reverse cumulative distribution function | Individuals with anti-S IgG lab result of at least 171 BAU (95% CI: 57, 519) are protected from infection | Anti-S IgG against Alpha: 171 BAU (95% CI: 57, 519) |

(*Continued*)

**Table 5.** (Continued)

| First author and publication year | Vaccine(s) investigated | Antibody assay and target, isotype | Primary outcome | Correlation | Statistical model used | Result and interpretation | Reported correlate of protection |
|---|---|---|---|---|---|---|---|
| **Gilbert, 2021** (Please see Table 4 for additional evidence) | Moderna | Lentivirus pseudoneutralization assay, cID50 | | | Causal inference approach using Cox regression | An estimated 68.5% (95% CI 58.5,78.4%) of VE was mediated by Day 29 cID50 titer | Not provided |
| | | Lentivirus pseudoneutralization assay, cID80 | | | Causal inference approach using Cox regression | An estimated 48.5% (95% CI 34.5, 62.4%) of VE was mediated by Day 29 cID80 titer | |

#-Assay results from each study were included for every antibody type (i.e.–anti-S, anti-N, anti-RBD) and isotype (i.e.–IgG, IgM, IgA) measured. In instances where more than one assay target was used to measure the same antibody target in the same study (i.e.–both PRNT and pseudoneutralization results, or anti-S results from two different assays), we included only one of these results. Full data extraction for every study can be provided upon request.

Definitions: anti-S = anti-spike, anti-N = anti-nucleocapsid, anti-RBD = anti-receptor binding domain, PRNT = plaque reduction neutralization test, LDT = laboratory-determined test, ELISA = enzyme-linked immunosorbent assay, OD = optical density, IRR = incidence rate ratio, HCW = health care worker, LTC = long term care, HCS = human convalescent sera, NAAT = nucleic acid amplification testing

standard, three studies [30, 31, 34] normalized antibody concentrations against convalescent sera used in their respective clinical trials, and reported antibody concentrations as a ratio of the antibody concentration/convalescent serum concentration. The remaining two studies [32, 33] provided results using the WHO IS.

Using different statistical methods, three studies [31–33] attempted to quantitate the contribution of antibodies to VE measures. Earle et al. incorporated data from seven vaccine clinical trials and reported that neutralizing antibodies accounted for 77.5% to 84.4% of VE [31]. Gilbert et al. focused on mRNA-1273 clinical trial data and reported that neutralizing antibodies accounted for 48.5% (95% CI 34.5, 62.4%) to 68.5% (95% CI 58.5, 78.4%) of VE [32]. This approach was also taken to estimate the effect of anti-S antibodies, with Earle and colleagues finding that anti-S antibody accounts for 91.3% to 94.2% (no CIs provided) of variation in efficacy [31]. Using data from individuals vaccinated with BNT162b2, mRNA-1273, ChAdOx1 nCoV-19 or Ad26.COV2.S, Goldblatt et al. reported that anti-S antibodies account for 68.6% to 97.4% (no CIs provided) of variation in efficacy [33].

Two studies estimated a SARS-CoV-2 threshold of protection. Goldblatt et al. used a random effects meta-analytic approach to calculate protective thresholds in WHO IS units for ancestral strain SARS-CoV-2 and Alpha (B.1.1.7) of 154 (95% CI 42, 559) and 171 (95% CI 57, 519) anti-S binding antibody units (BAU/mL), respectively. Khoury and colleagues used a protective neutralization classification model to estimate the antibody concentration resulting in 100% protection, which they estimated to be 28.6% (95% CI 19.2–29.2%) of the mean convalescent antibody level [34]. The authors also applied a logistic model to calculate the 50% protective neutralization level for symptomatic disease (the titre at which 50% of individuals are protected from symptomatic infection), which was found to be 20.2% (95% CI 14.4, 28.4) of the mean convalescent antibody level. This level corresponded to a neutralization titre of between 1:10 to 1:30 in most assays, which the authors estimate corresponds to 54 (95% CI 30–96) international units (IU)/ml. For severe disease, the 50% threshold was estimated to be only 3% (95% CI 0.71, 13.0%) of the mean convalescent level.

## Quality assessment

During quality assessment (S2 Table), we excluded studies that provided inadequate antibody measures or were missing sampling dates, data or laboratory methods details. Of the included studies, we noted that few reported antibody levels at 30–60 days post infection or vaccination or within 30 days of re-infection or breakthrough [20–22, 26, 28, 35, 37], the time periods which would provide the most insight on antibody levels.

## Discussion

Our systematic review found mixed evidence regarding a SARS-CoV-2 CoP, with a lack of standardization between laboratory methodology, assay targets, and sampling time points complicating comparisons and interpretation. Studies examining the relationship between antibody levels and VE presented high correlation coefficients, despite utilizing diverse data that included several vaccines and a variety of assays, VE endpoints and populations [30, 31, 33, 34]. The robust correlations despite data heterogeneity support the concept of an anti-S antibody or neutralizing antibody CoP. Furthermore, several studies that explored differences in GMTs between cases and non-cases [29, 32] or associations between antibody levels and viral load with infection incidence or risk [22, 29, 32, 39], found statistically significant differences and associations. Taken together, these aggregate data reports support an antibody target as a potential correlate. However, individual-level data provided contradictory findings. Individuals described in case reports experienced re-infection or breakthrough infection with considerable anti-S or neutralizing antibody levels pre-infection. Studies that attempted to estimate the contribution of antibody levels to VE measures [31–33] found that a substantial proportion of VE was not explained by antibody levels, suggesting that antibodies are only one component of protection. These findings echo SARS-CoV-2 vaccine trial data showing protection after one dose with very low levels of neutralizing antibodies, and suggest that cellular immunity or non-neutralizing antibodies may also play a role in protection [31, 41].

Our review of the literature indicates that a humoral SARS-CoV-2 CoP may be relative, such that antibodies reduce risk of infection but not eliminated it [4]. An analogous example is the influenza 50% protective dose, defined as the antibody concentration at which the risk of infection is reduced by half [3, 42]. Khoury and colleagues provided evidence for a relative correlate in calculating a "50% protective neutralization level" across vaccine studies, and found that lower antibody levels are required to prevent severe disease than to prevent symptomatic infection [34]. Our findings are also in line with real-world observations where SARS-CoV-2 breakthrough cases are often mild or asymptomatic, suggesting that while there is not adequate immunity to prevent infection, there is adequate immunity to prevent symptomatic or severe disease [43, 44]. Furthermore, since mRNA vaccines produce high antibody levels while viral vector vaccines result in robust cellular immunity, it is also possible that the CoP following vaccination may differ by vaccine product [33, 41].

Other data sources that were not eligible for inclusion in our review are supportive of a humoral CoP. Transfer of SARS-CoV-2 convalescent IgG to naïve rhesus macaques was found to be protective [45], and convalescent plasma and monoclonal antibody therapy have been used clinically [46, 47]. Although neither animal model nor passive transfer of immunity mimics the human immune response precisely, these data underscore the importance of humoral immunity for protection against SARS-CoV-2.

There were several limitations to the available literature for this systematic review. We included several case-reports, which generally provide a lower level of evidence and are prone to bias. The included studies used different laboratory assays and heterogeneity in targets. The WHO IS was seldom used, and the diversity of laboratory assays and results precluded a meta-

analysis of our data. To overcome the lack of calibration between laboratory assays, some studies normalized results against convalescent sera. However, since the humoral immune response to natural infection varies by age and disease severity [48], this method is not ideal. Most studies did not report which SARS-CoV-2 lineage. With the emergence of Omicron (B.1.1.529), the lack of Omicron-specific serological data prior to re-infection or breakthrough is unfortunate. Evidence based on *in vitro* neutralization assays suggests that, for immune responses to Omicron in individuals who have already been exposed to Ancestral SARS-CoV-2 antigens (whether through infection or vaccination), an Omicron CoP may be higher than for Ancestral SARS-CoV-2 or other VOCs, due to the reduced effectiveness of Ancestral antibodies for variant spike protein. To that point, Pfizer-BioNTech has reported a 25-fold reduction in neutralization titres against Omicron compared to Ancestral SARS-CoV-2 in individuals vaccinated with two doses of BNT162b2 [49]. Studies from South Africa and Germany report a reduction in neutralization up to 41-fold [50, 51], despite two or three doses of BNT162b2 or mRNA-1273 and previous infection. However, neutralization levels cannot be interpreted with regards to immunity in the absence of a CoP. This issue will be further complicated as the proportion of individuals with an Omicron-specific immune response due to infection, re-infection or breakthrough increases, especially if the clinical serology tools available for diagnostic purposes continue to use Ancestral SARS-CoV-2 antigens. Since a CoP will undoubtedly be variant-specific, continued study in this area is warranted as further variants are detected and vaccination policies evolve in response.

Our review did not examine the role of cellular immunity, which is a limitation because both animal models and human studies have suggested that cellular immunity is likely integral to protection [45]. Furthermore, the included studies focused on systemic immunity, which limits our ability to comment on mucosal antibodies, a known element of SARS-CoV-2 immunity [52]. Only three studies included in our review measured IgA levels in serum [16, 24, 37]. Since circulating IgA cannot be effectively transported into secretions [53], these studies cannot shed light on potential mucosal correlates of protection.

Our findings emphasize that further research into the role of humoral immunity, including non-neutralizing antibody, Fc effector functions and cellular and mucosal immunity is a priority, especially in the context of immune-evading variants like Omicron. The effect of lineage, vaccine product and the endpoint being measured (i.e. infection, symptomatic disease, severe disease) on the CoP are also essential questions. Currently, 40.5% of the world's population has not been vaccinated against SARS-CoV-2 [54]. The need to approve more vaccines is urgent, but placebo controlled trials have become difficult to perform [33]. A temporary CoP, even if imperfect, would allow us to break through this impasse by performing non-inferiority studies to authorize new vaccine products.

Taken together, our findings suggest that humoral immunity is an integral part of protection against SARS-CoV-2, and that an antibody target is the most likely immune marker for a SARS-CoV-2 CoP.

## Supporting information

**S1 Table. Full search strategy.**
(PDF)

**S2 Table. PRISMA reporting checklist.**
(DOCX)

**S3 Table. Quality appraisal of included manuscripts.**
(ZIP)

## Author Contributions

**Conceptualization:** Julie Perry, Melissa Richard-Greenblatt, Sarah A. Buchan, Manish Sadarangani, Shelly Bolotin.

**Data curation:** Julie Perry, Selma Osman, James Wright, Shelly Bolotin.

**Formal analysis:** Julie Perry, Selma Osman, James Wright, Shelly Bolotin.

**Funding acquisition:** Shelly Bolotin.

**Methodology:** Melissa Richard-Greenblatt, Sarah A. Buchan, Manish Sadarangani, Shelly Bolotin.

**Project administration:** Julie Perry.

**Supervision:** Shelly Bolotin.

**Writing – original draft:** Shelly Bolotin.

**Writing – review & editing:** Julie Perry, Selma Osman, James Wright, Melissa Richard-Greenblatt, Sarah A. Buchan, Manish Sadarangani, Shelly Bolotin.

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
