## [Decision Letter · Decision Letter 0]

9 Mar 2022

PONE-D-22-03976

Does a humoral correlate of protection exist for SARS-CoV-2? A systematic review

PLOS ONE

Dear Dr. Bolotin,

Thank you for submitting your manuscript to PLOS ONE. After careful consideration, we feel that it has merit but does not fully meet PLOS ONE’s publication criteria as it currently stands. Therefore, we invite you to submit a revised version of the manuscript that addresses the points raised during the review process.

During the revision process, please address the minor revisions recommended by the reviewers.

We look forward to receiving your revised manuscript.

Kind regards,

Victor C Huber

Academic Editor

PLOS ONE

“This work was supported by Public Health Ontario and funding from the Public Health Agency of Canada to SB.

NO”

“This work was supported by Public Health Ontario and funding from the Public Health Agency of Canada.”

“This work was supported by Public Health Ontario and funding from the Public Health Agency of Canada to SB.

NO”

“NO”

Reviewers' comments:

Reviewer's Responses to Questions

**Comments to the Author**

1. Is the manuscript technically sound, and do the data support the conclusions?

Reviewer #1: Yes

Reviewer #2: Yes

2. Has the statistical analysis been performed appropriately and rigorously? 

Reviewer #1: Yes

Reviewer #2: Yes

3. Have the authors made all data underlying the findings in their manuscript fully available?

Reviewer #1: Yes

Reviewer #2: Yes

4. Is the manuscript presented in an intelligible fashion and written in standard English?

Reviewer #1: Yes

Reviewer #2: Yes

5. Review Comments to the Author

Reviewer #1: The authors performed a literature review trying to determine the CoP for SARS-CoV-2 by compiling serological data on reinfections or vaccine breakthrough infections previously reported in publications or preprints. They found that while no absolute threshold for CoP can be determined up to this point, the correlate appears to be relative in a way that higher serological antibody levels in response to vaccination or infection yield greater protection against SARS-CoV-2 infection.

The manuscript is very well written, but its greatest strength lies not in its findings or conclusions which are fairly generic and previously reported on, but in the vast compendium of publications they found and assessed that have attempted correlating protection against SARS-CoV-2 with serological measures. The authors are well aware of the limitations, most of which stem from the fact that there is simply too much variables in the studies – the assays and units used, the types of vaccines, and the variants causing the re-infections or breakthrough infections. The authors allude to the fact that the CoP may be different between vaccine manufacturers and variants and no single answer can be found – that is certainly the case.

The authors also reported contradictory patterns from individual case studies regarding patterns between reinfection/breakthrough rates and prior antibody levels; this is not surprising as most case studies are different from the average (why they were deemed interesting as stand-alone cases) and should at most be weighted the same as a participant in a cohort study.

Overall, the publication has merit as the tables serve as a useful reference guide for SARS-CoV-2 researchers and the authors are well aware of the limitations of the study. I recommend the review for publication.

Reviewer #2: This paper is a review of serological correlates of protection for COVID-19 vaccines. The review is competent and complete. It reaches the conclusion that nearly everybody else has: that antibodies, particularly neutralizing, correlate with efficacy. It emphasizes that the correlate is relative, that is there is no level that corresponds to 100% efficacy. That conclusion is expected in view of published data and the fact that SARS-2 infection is primarily mucosal, similar to influenza, for which as the authors themselves say antibody is the correlate of protection although the protective level of antibody increases with titer but no absolute level is reached.. So the last line of the discussion “we do not have the tools to interpret serology with regards to protection” is silly, inasmuch as the authors describe how higher levels correlate with protection. Of course, cellular and Fc Effector immunity undoubtedly also have a role, but that does not gainsay the major role of antibody. It is also worth emphasizing that homologous levels have to be determined for each SARS-2 variant. On a trivial note, in line 162 “sera” is a plural noun and therefore “are”.

6. PLOS authors have the option to publish the peer review history of their article (what does this mean?). If published, this will include your full peer review and any attached files.

Reviewer #1: No

Reviewer #2: No

---

## [Author Response · Author response to Decision Letter 0]

24 Mar 2022

The manuscript has been reformatted according to PLOS style requirements.

“This work was supported by Public Health Ontario and funding from the Public Health Agency of Canada to SB.

NO”

A Role of Funder statement is now included in our cover letter

“This work was supported by Public Health Ontario and funding from the Public Health Agency of Canada.”

“This work was supported by Public Health Ontario and funding from the Public Health Agency of Canada to SB.

NO”

All funding-related statements have been removed from the text and included in the Cover letter and Funding statement. Please use the Funding Statement as previously submitted.

“NO”

Cover letter now includes conflict of interest disclosure, which has been removed from the text.

Reviewer comments (Responses in Bold/Italics)

Reviewer #1: The authors performed a literature review trying to determine the CoP for SARS-CoV-2 by compiling serological data on reinfections or vaccine breakthrough infections previously reported in publications or preprints. They found that while no absolute threshold for CoP can be determined up to this point, the correlate appears to be relative in a way that higher serological antibody levels in response to vaccination or infection yield greater protection against SARS-CoV-2 infection.

The manuscript is very well written, but its greatest strength lies not in its findings or conclusions which are fairly generic and previously reported on, but in the vast compendium of publications they found and assessed that have attempted correlating protection against SARS-CoV-2 with serological measures. The authors are well aware of the limitations, most of which stem from the fact that there is simply too much variables in the studies – the assays and units used, the types of vaccines, and the variants causing the re-infections or breakthrough infections. The authors allude to the fact that the CoP may be different between vaccine manufacturers and variants and no single answer can be found – that is certainly the case.

The authors also reported contradictory patterns from individual case studies regarding patterns between reinfection/breakthrough rates and prior antibody levels; this is not surprising as most case studies are different from the average (why they were deemed interesting as stand-alone cases) and should at most be weighted the same as a participant in a cohort study.

Overall, the publication has merit as the tables serve as a useful reference guide for SARS-CoV-2 researchers and the authors are well aware of the limitations of the study. I recommend the review for publication.

We thank the reviewer for their time and their expert appraisal of our manuscript.

Reviewer #2: This paper is a review of serological correlates of protection for COVID-19 vaccines. The review is competent and complete. It reaches the conclusion that nearly everybody else has: that antibodies, particularly neutralizing, correlate with efficacy. It emphasizes that the correlate is relative, that is there is no level that corresponds to 100% efficacy. That conclusion is expected in view of published data and the fact that SARS-2 infection is primarily mucosal, similar to influenza, for which as the authors themselves say antibody is the correlate of protection although the protective level of antibody increases with titer but no absolute level is reached.. So the last line of the discussion “we do not have the tools to interpret serology with regards to protection” is silly, inasmuch as the authors describe how higher levels correlate with protection. Of course, cellular and Fc Effector immunity undoubtedly also have a role, but that does not gainsay the major role of antibody. It is also worth emphasizing that homologous levels have to be determined for each SARS-2 variant. On a trivial note, in line 162 “sera” is a plural noun and therefore “are”.

We thank the reviewer for their time and their expert appraisal of our manuscript. We have removed the last line of the discussion, and also changed line 162 (now line 163) to reflect “sera” as plural. We have also added further emphasis to the variant-specific nature of a CoP in lines 281-283.

---

## [Editor Report · Decision Letter 1]

29 Mar 2022

Does a humoral correlate of protection exist for SARS-CoV-2? A systematic review

PONE-D-22-03976R1

Dear Dr. Bolotin,

We’re pleased to inform you that your manuscript has been judged scientifically suitable for publication and will be formally accepted for publication once it meets all outstanding technical requirements.

Kind regards,

Victor C Huber

Academic Editor

PLOS ONE
---

## [Editor Report · Acceptance letter]

31 Mar 2022

PONE-D-22-03976R1 

Does a humoral correlate of protection exist for SARS-CoV-2? A systematic review 

Dear Dr. Bolotin:

I'm pleased to inform you that your manuscript has been deemed suitable for publication in PLOS ONE. Congratulations! Your manuscript is now with our production department. 

Kind regards, 

on behalf of

Dr. Victor C Huber 

Academic Editor

PLOS ONE